# From Modeling to Sensing of Micro-Doppler in Radio Communications

**Louis Morge-Rollet [1,\*]**, **Denis Le Jeune [1]**, **Frédéric Le Roy [1]**, **Charles Canaff [1]** and **Roland Gautier [2]**

1    ENSTA Bretagne, Lab-STICC, CNRS, UMR 6285, F-29200 Brest, France
2    Univ Brest, Lab-STICC, CNRS, UMR 6285, F-29200 Brest, France
\*    Correspondence: louis.morge-rollet@ensta-bretagne.org

**Abstract:** The Doppler effect in radio systems has been widely explored by the radio communication community. However, these studies have been limited to simple motion such as linear translation. This paper presents a model for the Doppler modulation effect, i.e., the effect of complex movement on the received signal, using a geometrical approach. Particularly, we focused on studying micro-Doppler in radio communications produced by vibrations. Exploiting this phenomenon would allow the performance of passive micro-Doppler effect sensing based on communication. In this paper, we also propose signal processing techniques to detect the presence of the micro-Doppler effect and to estimate its parameters. Then, we present some experiments which highlight the micro-Doppler effect in a radio communication context. Finally, the end of the paper discusses some potential applications that exploit this phenomenon.

**Keywords:** micro-Doppler; passive micro-Doppler sensing; communication signal cancellation; signal detection; signal parameter estimation; RF fingerprinting

## 1. Introduction

From a wireless communications perspective, the Doppler effect is considered to model movement effects in propagation environment [1]. It is used to model the effect of a relative motion between a transmitter and a receiver. The general model for Doppler effect is to consider a line-of-sight (LOS) configuration, i.e., a unique direct path between the transmitter and the receiver. In this configuration, it is considered to produce a frequency shift $e^{j2\pi f_d t}$ of the baseband signal (with $f_d$ the Doppler shift). However, channel propagation is usually more complex than a frequency shift due to the Doppler effect and multipath configuration. Thus, some channel models rely on both aspects, such as multipath under differential Doppler [2] or statistical channel models [3] using Doppler fading [4]. In some situations, the Doppler effect is considered to dilate or compress the received signal when the speed of the wave is of the same order of magnitude as the transmitter speed. This dilation/compression phenomenon is particularly present in acoustic communications [5,6].

There are many other applications of the Doppler effect in the present day, particularly in the field of remote sensing techniques. Its exploitation was historically used in astronomy to measure the radial speed [7] or temperature of a star. However, radar is one of the fields that most exploits the Doppler effect. Indeed, it is classically used in the field of radar to measure the speed of a target, for example for a road radar or for a heading direction estimation. However, there are several other uses of Doppler effect in radar, such as micro-Doppler [8,9]. Micro-Doppler are generated by micro-motions, such as vibrations, on the returned signal and can be used for several applications such as drone detection [10] and motion sensing [11]. Other domains also take advantage of the Doppler effect for remote sensing purposes, for example, for laser vibrometry [12] or for medical applications such as echoDoppler.

Conversely, there are few references where the Doppler effect is used for remote sensing purposes in radio communication systems such as, for example, [13] on drone detection and [14] on Doppler-assisted wireless communication. Active RF sensing techniques have also been studied for drone detection exploiting micro-Doppler signatures in non-cooperative scheme with an RF single tone as an illuminator [15]. Similar techniques are referred to in the literature as passive radar and exploit communication signals from an illuminator such as a base station [16]. Particularly, several approaches in passive radar have focused on micro-Doppler effect extraction and analysis from backscattering signal using communication signals such as GNSS [17] or OFDM [18]. Conversely, our approach exploits the micro-Doppler effect produced by vibrations at the antenna of the transmitter instead of exploiting an illuminator. It can be noted that both micro-Doppler signatures provide complementary information to characterize a target.

We will model the effect of complex movement on the received signal for wireless communication systems from the perspectives of performing remote sensing tasks based on communication itself. Specifically, this work presents the Doppler modulation phenomenon in radio communications both theoretically and experimentally and explains how to extract and exploit it for future applications. The contributions of this paper are the following: (1) modeling the effect of a complex movement on the received signal; (2) provision of tools for extracting Doppler modulation from communication signals and detecting and estimating the micro-Doppler effect; and (3) illustration of the phenomenon using real-world experiments.

Section 2 is about modeling of the Doppler modulation effect. Section 3 deals with communication signal cancellation for Doppler modulation extraction. Section 4 describes the signal model of Doppler modulation and the detection and estimation algorithms of the Doppler modulation effect. Finally, Section 5 demonstrates potential use cases of the phenomenon among simulation, vibrating object detection, RF fingerprinting, and vibration analysis.

## 2. Modeling of the Doppler Modulation Effect

We will model the Doppler modulation effect using a geometrical model for communication systems. Particularly, we will focus on studying the micro-Doppler effect (the modulation of micro-movement such as vibration) on the received baseband signal. Furthermore, we also present a testbed and an experiment that highlight the micro-Doppler effect on a simple transmitted signal, i.e., a signal tone.

### 2.1. Reminders

Classically, the Doppler effect is represented by:

$$f_{RX} = \frac{c - v_{RX}}{c - v_{TX}} f_{TX} \tag{1}$$

$$= \frac{1 - \frac{v_{RX}}{c}}{1 - \frac{v_{TX}}{c}} f_{TX} \tag{2}$$

with:

- $c$ the speed of light;
- $f_{TX}$ the source frequency;
- $f_{RX}$ the received frequency;
- $v_{TX}$ the source speed;
- $v_{RX}$ the receiver speed.

In radio communications, the Doppler effect can also produce a dilation/compression of the emitted signal bandwidth called Doppler spread. However, this bandwidth dilation/compression is usually neglected because the Doppler spread is small in comparison to the bandwidth [1]. Thus, the Doppler effect is considered to simply produce a frequency shift of the emitted signal.

In this paper, we consider the case of narrowband communication signals. The narrowband signal conditions required that signal bandwidth $B$ is negligible in comparison to a central frequency $f_c$ and the corresponding model is described as follows [19,20] :

$$s(t) = a(t)\cos(2\pi f_c t + \phi(t)) \tag{3}$$

with:

- $a(t)$ an amplitude modulation;
- $\phi(t)$ a phase modulation.

Its baseband representation:

$$\tilde{s}(t) = a(t)e^{j\phi(t)} \tag{4}$$

and its analytic signal:

$$s_a(t) = \tilde{s}(t)e^{2\pi f_c t} \tag{5}$$

In this article, we consider $\mathbb{E}(|\tilde{s}(t)|^2) = 1$, i.e. the baseband signal power is unitary. However, the transmitted power $P_t$ can be different from 1, reflecting the signal amplification process of the emitter.

### 2.2. Geometrical Modeling

This demonstration use a geometrical model of the movement between the transmitter and the receiver presented in Figure 1 (inspired by the work of Lyonnet [5]). This method is based on the ray tracing theory, i.e., a modeling theory widely used in radio communication for far-field propagation (see [1]). The rays are the trajectories, which are orthogonal to the wavefronts (constant phase surface) and thus corresponding to the direction of the wave propagation. The ray tracing theory is a good approximation of a far-field wave propagation in the space. We will make the assumption of a unique direct path, but the demonstration can be extended to multipath configurations using virtual sources [5].

We will introduce the angle $\alpha$, which is the angle between the two vectors $\vec{l}_i(0)$ and $\vec{x}(t)$. Furthermore, we will suppose the vector $\vec{x}(t) = x(t)\vec{x}$, where $x(t)$ is a real function (that can be either positive or negative depending on time) and $\vec{x}$ a unitary vector oriented as in Figure 1.

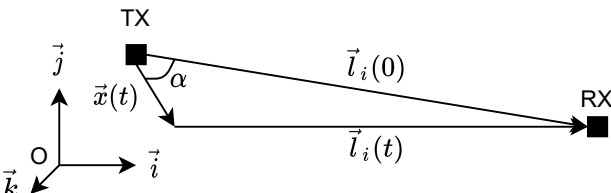

**Figure 1.** Geometrical model.

The signal is sent by the transmitter (TX) using an isotropic antenna and is also received by the receiver (RX) using an isotropic antenna, so the received signal expression is the following:

$$r(t) = A(t - \tau_i(t))s(t - \tau_i(t)) \tag{6}$$

with:

- $\tau_i(t)$: the signal delay depending on time with $\tau_i(t) = \frac{\|\vec{l}_i(t)\|}{c}$;
- $A(t)$: a loss term due to propagation depending on $\|\vec{l}_i(t)\|$.

To demonstrate the effect of movement $\vec{x}(t)$ on the received signal $s(t)$, we make following hypothesis:

- **Hypothesis 1**: $\|\vec{x}(t)\| << \|\vec{l}_i(0)\|$, i.e., the transmitter movement $\|\vec{x}(t)\|$ is negligible compared to the initial distance $\|\vec{l}_i(0)\|$ between the transmitter and receiver during signal observation time $T$.

- **Hypothesis 2**: $\left\|\vec{l_i}(t)\right\| \approx \left\|\vec{l_i}(t - \tau_i(t))\right\|$, i.e., the distance between the transmitter and receiver at time $t - \tau_i(t)$ is approximately the same as at time $t$.
- **Hypothesis 3**: Depending of the movement $\vec{x}(t)$, one of the two following sub-hypotheses should be chosen:
  - **Hypothesis 3a**: In case of an arbitrary movement $\vec{x}(t)$, the loss term $A(t)$ is considered constant, i.e., the loss term is considered independent of $\|\vec{l_i}(t)\|$ (stronger hypothesis but more general);
  - **Hypothesis 3b**: In case of a small movement $\vec{x}(t)$ ($k\|\vec{x}(t)\| << 1$), $k\|\vec{l_i}(0)\| >> 1$, i.e., the product of the wavenumber $k = \frac{2\pi}{\lambda}$ and the initial distance $\|\vec{l_i}(0)\|$ is much greater than 1 (weaker hypothesis but less general).

Considering the hypotheses previously introduced, we obtain the following expression for received signal (Appendix A):

$$r(t) = Aa(t - \tau_0 + \frac{\langle \vec{x}(t), \vec{l_{i0}} \rangle}{c})$$

$$\cos(2\pi f_c t + k\langle \vec{x}(t), \vec{l_{i0}} \rangle + \phi(t - \tau_0 + \frac{\langle \vec{x}(t), \vec{l_{i0}} \rangle}{c}) - \phi_0) \quad (7)$$

with:

- $A = A(0)$: a constant loss term due to propagation;
- $\tau_0 = \frac{\|\vec{l_i}(0)\|}{c}$: the initial delay;
- $\vec{l_{i0}} = \frac{\vec{l_i}(0)}{\|\vec{l_i}(0)\|}$: the unitary vector of $\vec{l_{i0}}$;
- $\phi_0 = 2\pi f_c \tau_0$: the initial phase.

Furthermore, its corresponding baseband signal:

$$\tilde{r}(t) = Aa(t - \tau_0 + \frac{\langle \vec{x}(t), \vec{l_{i0}} \rangle}{c})e^{j(k\langle \vec{x}(t), \vec{l_{i0}} \rangle + \phi(t - \tau_0 + \frac{\langle \vec{x}(t), \vec{l_{i0}} \rangle}{c}))} \quad (8)$$

$$= A\tilde{s}(t - \tau_0 + \frac{\langle \vec{x}(t), \vec{l_{i0}} \rangle}{c})e^{-j\phi_0}e^{jk\langle \vec{x}(t), \vec{l_{i0}} \rangle} \quad (9)$$

For the rest of the article, we will consider that we can decompose the movement $\vec{x}(t) = \vec{v}t + \vec{m}(t)$ as sum of a linear translation depending of vector $\vec{v}$ and a micro-movement $\vec{m}(t)$ due to vibration. It can be noted that, for simplicity, we consider both vectors depending of the same direction, but it is possible to extend the demonstration to a sum of movements $\vec{x}(t) = \sum_{i=1}^{N} \vec{x_i}(t)$ with different directions. Thus, we will then write $\langle \vec{x}(t), \vec{l_{i0}} \rangle = (vt + m(t))\cos(\alpha)$ and the received baseband signal will become:

$$\tilde{r}(t) = A\tilde{s}(\xi t - \tau_0 + \frac{m(t)}{c}\cos(\alpha))e^{-j\phi_0}e^{jk\langle \vec{x}(t), \vec{l_{i0}} \rangle} \quad (10)$$

with $\xi = (1 + \frac{v}{c}\cos(\alpha))$, the dilation/compression factor.

We can observe that the geometrical modeling allows for modeling of the Doppler modulation term $e^{jk\langle \vec{x}(t), \vec{l_{i0}} \rangle}$ but also the effect of a complex movement $\vec{x}(t)$ on the transmitted baseband signal. If we consider $\vec{x}(t) = \vec{v}t + \vec{m}(t)$, we can observe that the Doppler modulation term can be decomposed of the product of a micro-Doppler term $e^{jkm(t)\cos(\alpha)}$ and a Doppler shift term $e^{j2\pi f_d t}$ with $f_d = \frac{v}{c}f_c\cos(\alpha)$. Furthermore, the received baseband signal is dilated/compressed by a factor $\xi = 1 + \frac{v}{c}\cos(\alpha)$ due to the linear translation, but it also undergoes a Doppler micro-jitter phenomenon $\frac{m(t)}{c}\cos(\alpha)$ (similar to a phase noise term) due to the micro-movement $\vec{m}(t)$. As Tse explained in their book [1], the dilation/compression phenomenon due to the translation can be ignored in practice, because the Doppler spread ($f_d = \frac{v}{c}f_c\cos(\alpha)$) is small (of the order of tens to hundred of Hz in radio communications) compared to the bandwidth B. On the one hand, if we consider that

the maximum speed is 300 km/h, which is widely used for telecommunication standards such as WiFi, LTE, and even GSM, we obtain a maximum Doppler offset $f_d \approx 680$ Hz for 2.45 GHz ISM band, which is equivalent to 0.28 ppm. On the other hand, the Doppler micro-jitter effect due to the micro-movement $m(t)$ can also be characterized. If we consider that the micro-movement is periodic, the resulting phase noise term is a periodic jitter.

Indeed, in this paper, we consider the case of vibrations that are considered often periodic, especially in rotating machines [21]. Usually, the micro-jitter is negligible in comparison to the phase noise of an oscillator because the vibration magnitude is in the order of a millimeter.

As a first approximation of the received signal, we will neglect the Doppler micro-jitter on the signal:

$$\tilde{r}(t) \approx A\tilde{s}(\xi t - \tau_0)e^{-j\phi_0}e^{jk\langle \vec{x}(t), \vec{l_{i0}}\rangle} \tag{11}$$

The dilation/compression can also be neglected [1]. For this reason, we can also neglect the dilation/compression and the following signal:

$$\tilde{r}(t) \approx A\tilde{s}(t - \tau_0)e^{-j\phi_0}e^{jk\langle \vec{x}(t), \vec{l_{i0}}\rangle} \tag{12}$$

More generally, if we consider that $\langle \vec{x}(t), \vec{l_{i0}}\rangle << c$, the Doppler modulation effect on the received baseband signal can be expressed similarly as in Equation (12).

It is important to note that the geometrical modeling can be used from a radar point of view. Indeed, by redesigning Figure 1 to take into account a radar/target configuration (and not a receiver/transmitter configuration) and by redefining the delay $\tau_i(t) = \frac{2\|\vec{l_i}(t)\|}{c}$ to take account the signal backscattering and not a direct propagation, we can obtain the modeling of the movement's effect of the target on the received signal. This type of modeling is similar to the one proposed by V. Chen et al. in [8,9,11] for micro-Doppler effect in radar, although our modeling allows for the expression of the Doppler dilation and the Doppler micro-jitter phenomenon on baseband signal.

### 2.3. Testbed and Experiments

### 2.3.1. Testbed, Hypothesis and Formalization

We created a testbed (see Figure 2) to highlight the Doppler modulation phenomenon. In this study, we considered only the micro-Doppler modulation (due to micro-movements such as vibrations). This choice was justified by the decomposition of the movement $\vec{x}(t) = \vec{v}t + \vec{m}(t)$ and by the fact that the effect of linear translation has already been explored and theorized by the radio communication community. In this section, we will restrict the experiment to the simplest transmitted signal, i.e., a sinusoid (also called a single tone). The configuration of our experiment is the following:

- Hypothesis:
  - Isotropic antenna transmitting a sinusoid:
    $s(t) = \cos(2\pi f_{em}t + \Phi_{em})$;
  - The micro-movement is a sinusoid:
    $m(t) = A_v \sin(2\pi f_0 t + \Phi_0)$;
  - Resulting baseband signal (Jacobi–Anger expansion):

$$\tilde{r}(t) = Ae^{j(\beta \sin(2\pi f_0 t + \Phi_0) + \Phi)} \ (\beta = \frac{2\pi A_v}{\lambda}) \tag{13}$$

$$= Ae^{j\Phi} \sum_{n=-\infty}^{+\infty} (e^{j\Phi_0})^n J_n(\beta)e^{j2\pi n f_0 t} \tag{14}$$

- Transmitting system:
  - Sinusoid generator: ANRITSU MG3692B;
  - Transmitting antenna: Ettus VERT 2450;
  - Vibration generator: WOVELOT 037606, nominal voltage 1.5 V;

- IMU (inertial measurement unit): SparkFun 9DoF Razor IMU M0.
- Receiving system:
  - Signal recorder/signal analyzer: Signal Hound BB60C;
  - Receiving antenna: Ettus VERT 2450.
- Testing parameters:
  - Frequency: 2.45 GHz (affecting $\lambda$);
  - Voltage of vibration generator: 1.5 V (affecting $A_v$ and $f_0$);
  - Distance: 2 m (affecting $A$).

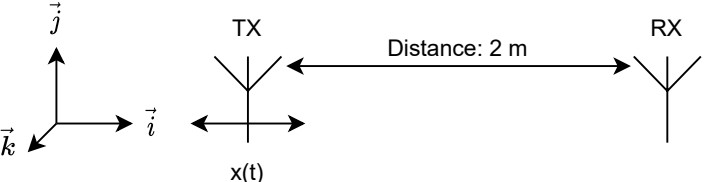

**Figure 2.** Experiment setup.

### 2.3.2. Results

Figure 3 presents the results of the previously described experiment. On the one hand, Figure 3a corresponds to the spectrum of the synchronized received baseband signal, i.e., the frequency offset has been corrected. On the other hand, Figure 3b corresponds to the spectrum of the IMU signal measuring the vibration at transmitting antenna. The theoretical model predicts that the harmonics of the spectrum of a received baseband signal are equally spaced with the value $f_0$. We can observe that this spacing is present in the spectrum and we can also observe that the spacing of the tones is the same as the vibration frequency ($f_0 = 122$ Hz) measured by the IMU (placed on the transmitting antenna). However, it can be noted that the magnitudes are not the same as the model prediction, probably because the antenna used is not isotropic (theoretical model) and the vibration (Figure 3) is not a pure sinusoid but has several harmonics. Indeed, the vibration is probably closer than a squared vibration shape due to the technology employed in the vibration generator (vibrator).

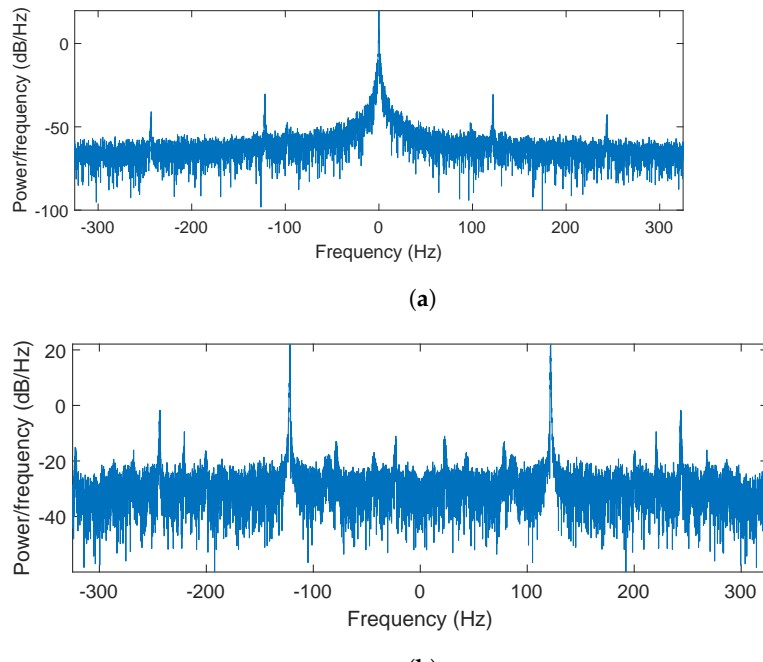

**Figure 3.** (**a**) Spectral density of synchronized baseband signal. (**b**) Spectral density of IMU signal.

### 3. Communication Signal Cancellation

In contrast to the radar domain where the emitted signal is known, in radio communications the received signal is unknown, except for the modulation scheme and protocol. So, it is critical to suppress this source of randomness (transmitted data) to better exploit micro-Doppler for providing useful information about an RF emitter. In this section, we propose a pre-processing step allowing to remove communication signal designed for M-ary phase-shift keying (MPSK) modulations. It allows the extraction of frequency offset and/or Doppler modulation terms from received baseband signal. We present the theoretical point of view of this approach and illustrate it with practical experiment. In the rest of the paper, to simplify the notations, we will consider $\langle \vec{x}(t), \vec{l_{i0}} \rangle = x(t)$. We will also neglect the propagation delay $\tau_0$ and initial phase $\phi_0$. Thus, the received baseband signal with additive white Gaussian noise $\tilde{n}(t)$ can be written as follows:

$$\tilde{r}(t) = A\tilde{s}(t)e^{jkx(t)} + \tilde{n}(t) \tag{15}$$

#### 3.1. Theoretical Explanations

In the previous section, we saw that it is possible to model the Doppler modulation effect on a received baseband signal using Equation (12). The main problem is that for very small movement, particularly for micro-movement such as vibration, it is difficult to measure directly the Doppler modulation on received baseband signal, either in the time or frequency domains. On the one hand, the Doppler modulation is similar to phase modulation, and due to the small amplitude of the micro-movement, analyzing it from the received baseband signal in the time domain it is extremely complex. On the other hand, the effect of the micro-movement on the received spectrum is completely hidden by the communication signal because both are convolved in the frequency domain.

The idea developed in this section is to perform some pre-processing on the received baseband signal to suppress the communication signal $\tilde{s}(t)$ and thus to facilitate the study of the Doppler modulation term. In this paper, we will consider the case of phase modulation (M-PSK). Thus, the following signal model can be expressed as follows:

$$\tilde{r}(t) = A\tilde{s}(t)e^{jkx(t)} + \tilde{n}(t) \tag{16}$$

with:

- $A$: the amplitude of the signal;
- $\tilde{s}(t) = \sum_{k=-\infty}^{+\infty} a_k \pi(\frac{t-kT_0}{T_0})$ the MPSK signal;
- $a_k = (e^{j\frac{2i\pi}{M}})_{i\sim U(\llbracket 0\,;\,M-1\rrbracket)}$;
- $M$: the modulation order;
- $\pi(t)$: the rectangular function;
- $T_0$: the symbol period;
- $\tilde{n}(t) \sim \mathcal{CN}(0,1)$: an additive white Gaussian complex noise.

A well-known method for the synchronization of M-PSK modulations (with rectangular function as pulse shaping filter) is the suppression of the modulation on a received signal by powering the received signal by a factor M (see Glavieux [22]):

$$\tilde{s}(t)^M = (\sum_{k=-\infty}^{+\infty} a_k \pi(\frac{t-kT_0}{T_0}))^M \tag{17}$$

$$= \sum_{k=-\infty}^{+\infty} a_k^M \pi(\frac{t-kT_0}{T_0}) \tag{18}$$

$$= \sum_{k=-\infty}^{+\infty} \pi(\frac{t-kT_0}{T_0}) \tag{19}$$

$$= 1 \tag{20}$$

This processing usually allows for the suppression of the modulation to estimate the frequency offset between the transmitter and the receiver. Our method of communication signal cancellation will be based on this pre-processing step, allowing suppression of the modulation while maintaining a term linked to the Doppler modulation effect ($e^{jkx(t)} \rightarrow e^{jkMx(t)}$). This pre-processing hence gives the following signal:

$$\tilde{r}(t)^M = (\tilde{s}(t)e^{jkx(t)} + \tilde{n}(t))^M \tag{21}$$

Using Newton's binomial theorem, we can decompose the signal similar to a sum of different terms:

$$\tilde{r}(t)^M = \sum_{m=0}^{M} \binom{M}{m} A^m \tilde{s}(t)^m e^{jkmx(t)} \tilde{n}(t)^{M-m} \tag{22}$$

If the SNR is relatively high, it is possible to keep only the first main terms of the previous expression (see Appendix B):

$$\tilde{r}(t)^M \approx A^M e^{jkMx(t)} + M A^{M-1} \tilde{s}(t)^{(M-1)} e^{jk(M-1)x(t)} \tilde{n}(t) \tag{23}$$

$$\approx A^M e^{jkMx(t)} + \tilde{n}_1(t) \tag{24}$$

Equation (24) contains two terms, the first corresponds to the modified Doppler modulation term $A^M e^{jkMx(t)}$ and the second one $\tilde{n}_1(t)$ can be assumed as an additive complex white noise (see Appendix B).

In summary, this pre-processing consists of powering the received signal by a factor M corresponding to the modulation order and allowing the suppression of the modulation ($\tilde{s}(t)^M = 1$) while keeping the Doppler modulation term (modified by the power term M). It is important to note that this processing is robust to a potential frequency offset $\Delta f$ between the transmitter and the receiver, because the resulting processed signal will contain the modified Doppler modulation term multiplied by a term function of the modified Doppler shift $e^{j2\pi\Delta ft} \rightarrow e^{j2\pi M\Delta ft}$.

### 3.2. Testbed and Experiment

#### 3.2.1. Testbed, Hypothesis and Formalization

We used a configuration inspired from the previous testbed (see Figure 2) to highlight our method of canceling communication signal. Specifically, we confined ourselves to the study of micro-Doppler (due to micro-movement such as vibration). This choice has been justified in the previous section. Furthermore, we will restrict the present experiment to the 2-PSK communication signal. The configuration of our experiment is the following:

- Hypothesis:
    - Isotropic antenna transmitting a 2-PSK:
      $\tilde{s}(t) = \sum_{k=-\infty}^{+\infty} a_k \pi(\frac{t-kT_0}{T_0})$
    - The micro-movement is a sinusoid:
      $m(t) = A_v \sin(2\pi f_0 t + \Phi_0)$
    - Resulting baseband signal:

$$\tilde{r}(t) = A^2 \tilde{s}(t)^2 e^{j\frac{4\pi A_v \sin(w_v t)}{\lambda}} + \tilde{n}_1(t) \tag{25}$$

- Transmitting system:
    - Signal generator: ANRITSU MS2830A;
    - Transmitting antenna: Ettus VERT 2450;
    - Vibration generator: WOVELOT 178037606, nominal voltage 1.5V.
- Receiving system:
    - Signal recorder/Signal Analyzer: Signal Hound BB60C;

 –   Receiving antenna: Ettus VERT 2450.
- Testing parameters:
   –   Frequency: 2.45 GHz;
   –   Voltage of vibration generator: 1.5 V;
   –   Distance: 2 m;
   –   Modulation: BPSK (rectangular pulse shaping).

### 3.2.2. Results

An example of the obtained signal is shown in Figure 4 for a BPSK (2-PSK) modulation with a rectangular function as pulse shaping filter. We can see in the upper subfigures (Figure 4a,b) the spectra obtained with a reference signal (i.e., a single tone). Contrarily, the lower subfigures (Figure 4c,d) show the spectra obtained with the pre-processing on BPSK . In parallel, the left subfigures (Figure 4a,c) correspond to configurations where the vibrator is off, and the right subfigures (Figure 4b,d) correspond to configurations where the vibrator is on. In the right subfigures, we can observe the presence of the micro-Doppler effect on the spectra (Figure 4b,d), contrasting Figure 4a,c and indicating the presence of vibration for both the single tone and pre-processed BPSK signal.

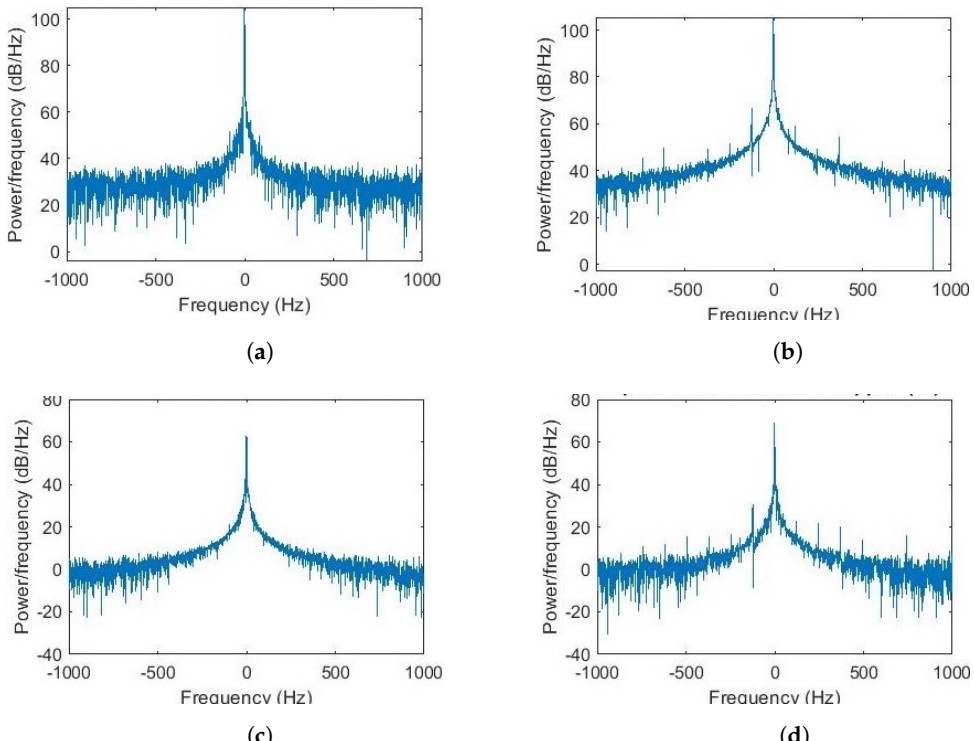

**Figure 4.** (**a**) Spectral density of single tone without Doppler. (**b**) Spectral density of single tone with Doppler. (**c**) Spectral density of BPSK without Doppler ($x^2$). (**d**) Spectral density of BPSK with Doppler ($x^2$).

## 4. Signal Model, Detection and Estimation

In this section, we propose a signal model derived from the decomposition of the movement $\vec{x}(t) = \vec{v}t + \vec{m}(t)$ used in the previous sections. Furthermore, we propose a binary hypothesis testing approach based on cyclostationary properties of the signal and an algorithm that allows for the estimation of different parameters of the signal. We also present simulation results for the detection and estimation.

*4.1. Signal Model of Doppler Modulation*

In signal processing, a signal model is a mathematical modeling of signals of interest such as the harmonics model for the MUSIC algorithm [23]. The proposed signal model aims to describe the Doppler modulation term $e^{jkx(t)}$. This term corresponds either to a received signal if the transmitted signal is a sinusoid or to a processed received signal in the case of M-PSK modulation using technique present in Section 3. Furthermore, we also model a frequency offset $\Delta f$ corresponding to the frequency difference between the transmitter and the receiver and an additive white Gaussian complex noise. In this part, we consider the amplitude term $A$ as unitary. It can be interpret as a normalization of the received signal by a factor $A$. The corresponding signal model is as follows:

$$\tilde{y}(t) = e^{jkx(t)}e^{j2\pi\Delta ft} + \tilde{n}(t) \tag{26}$$

$$= e^{jkm(t)}e^{j2\pi f_1 t} + \tilde{n}(t) \tag{27}$$

with:

- $x(t) = vt + m(t)$,
- $f_1 = \Delta f + f_d$: the total frequency offset taking into account the Doppler shift $f_d = \frac{v}{c}f_c$ and the frequency offset $\Delta f$ between transmitter and receiver;
- $\tilde{n}(t) \sim \mathcal{CN}(0, \sigma^2)$: an additive white Gaussian complex noise.

We consider the micro-movement $m(t)$ as periodic of frequency $f_0$, as it is common to consider a vibration as periodic in the literature [21]. We can decompose the micro-Doppler term using the Fourier series as follows:

$$e^{jkm(t)} = \sum_{n=-\infty}^{+\infty} a_n e^{j2\pi f_0 t} \tag{28}$$

with:

- $a_n = \int_{-\infty}^{+\infty} e^{jkm(t)}e^{-j2\pi n f_0 t}dt$: a Fourier series coefficient.

We obtain the following expression, the so-called theoretical model:

$$\tilde{y}(t) = (\sum_{n=-\infty}^{+\infty} a_n e^{j2\pi n f_0 t})e^{j2\pi f_1 t} + \tilde{n}(t) \tag{29}$$

$$= \sum_{n=-\infty}^{+\infty} a_n e^{j2\pi(n f_0 + f_1)t} + \tilde{n}(t) \tag{30}$$

The associated spectrum:

$$\tilde{Y}(f) = \sum_{n=-\infty}^{+\infty} a_n \, \delta(f - (n f_0 + f_1)) + \tilde{N}(f) \tag{31}$$

with:

- $\tilde{N}(f)$: the spectrum of the additive white Gaussian complex noise.

It can be noted that a subcase of this signal model has been already explained in [8] for sinusoidal vibration in a radar context.

*4.2. Detection Method*

4.2.1. Theoretical Explanations

In this work, we aim to detect micro-movement using a hypothesis test. As we can see in Figure 4, when the vibrator is off (subfigures Figure 4a,c), we obtain an exponential term similar to the frequency offset ($H_0$) between the transmitter and the receiver; however, when the vibrator is on (subfigures Figure 4b,d), we obtain the modulation term ($H_1$) described by the theoretical model.

We can then define two hypotheses for binary detection test:

$$H_0 \,:\, ae^{j2\pi f_1 t} + \tilde{n}(t)$$

$$H_1 \,:\, \sum_{n=-\infty}^{+\infty} a_n e^{j2\pi(nf_0 + f_1)t} + \tilde{n}(t) \tag{32}$$

The micro-Doppler term $e^{jkm(t)}$ can be decomposed using Fourier series due to periodic properties. So, it has also cyclostationary properties because periodic signals are part of a cyclostationary process [21]. We will then define a binary hypothesis testing approach (see Figure 5) that exploits the cyclostationary properties of the signal to detect the presence of micro-Doppler.

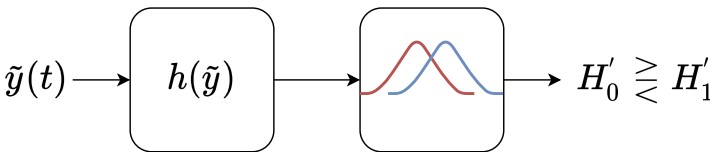

**Figure 5.** Detection process including transformation step.

Our detection test is based on a modified version of the binary hypothesis test [24] used for modulation detection in additive white Gaussian complex noise (see Appendix C). Note that [25] seems to extend this detection test for cyclostationary signals. Similar to them, our binary hypothesis testing approach use the cyclic frequency domain profile (CDP):

$$I(\alpha) = max_f |C_x^\alpha(f)| \tag{33}$$

with:

- $C_x^\alpha(f)$: the spectral coherence;
- $\alpha$: the cyclic frequency (here equivalent to $f_0$).

The alternative hypothesis $H_1$ (Equation (32)) is not cyclostationary due to the frequency offset term $e^{j2\pi f_1 t}$. Before performing the hypothesis test, we apply a transformation $h(\tilde{y})$ called synchronization on the incoming signal and presented in Equation (34). The associated hypotheses derived from this transformation are presented in Equation (35). Note that the resulting alternative hypothesis $H_1'$ now has cyclostationary properties.

The transformation $h(\tilde{y})$ is:

$$h(\tilde{y}) = \tilde{y}(t)e^{-j2\pi \hat{f}_1 t} \tag{34}$$

with:

- $\hat{f}_1$: the estimated frequency offset using Fourier transform as mentioned by [26].

The associated hypotheses:

$$H_0 \overset{h(\tilde{y})}{\to} H_0' : a + \tilde{n}'(t)$$

$$H_1 \overset{h(\tilde{y})}{\to} H_1' : \sum_{n=-\infty}^{+\infty} a_n e^{j2\pi nf_0 t} + \tilde{n}'(t) \tag{35}$$

with:

- $n'(t) = n(t)e^{-j2\pi \hat{f}_1 t}$ assumed to be an additive white Gaussian complex noise.

Our modified test is based on the following statistic:

$$C_I = \frac{I(\alpha)}{\sqrt{\frac{1}{N} \sum_{\alpha=\alpha_{min}}^{\alpha_{max}} I(\alpha)^2}} \tag{36}$$

Furthermore, a specific threshold computed for the null hypothesis:

$$C_{TH} = \frac{max(I(\alpha))}{\sqrt{\frac{1}{N} \sum_{\alpha=\alpha_{min}}^{\alpha_{max}} I(\alpha)^2}} \tag{37}$$

As mentioned in [24,25], $C_{TH}$ is a random variable due to random noise. The false alarm rate can be obtained by probability function estimation (see Appendix C).

The binary hypothesis testing is performed as follows:

$$\begin{aligned} C_I \leq C_{TH} &: Declare\ H_0 \\ C_I > C_{TH} &: Declare\ H_1 \end{aligned} \tag{38}$$

### 4.2.2. Results

One hundred simulations have been realized to compare performances over different observation times $T$ and different SNR values with sampling frequency $F_s = 1000$ Hz (see Figure 6). The signal used to model the micro-Doppler is a sinusoidal frequency-modulated (SFM) signal with a frequency offset component and an additive white Gaussian complex noise. The motivation to use an SFM signal is mainly because it is part of the signal model described by Equation (30). Furthermore, throughout this study we have considered a sinusoidal vibration for micro-movement. Finally, we used simulation because it is more convenient for the experimental data and makes the experiment controllable and reproducible. Thus, the signal used for simulation is the following:

$$\tilde{s}(t) = e^{j\beta \sin(2\pi f_0 t)} e^{j2\pi f_1 t} + \tilde{n}(t) \tag{39}$$

with:

- $\beta = 0.1$: the modulation index;
- $f_0 \in [30, 150]$ Hz: the periodic frequency;
- $f_1 \in [-50, 50]$ Hz: the frequency offset;
- $\tilde{n}(t)$: the additive white Gaussian complex noise.

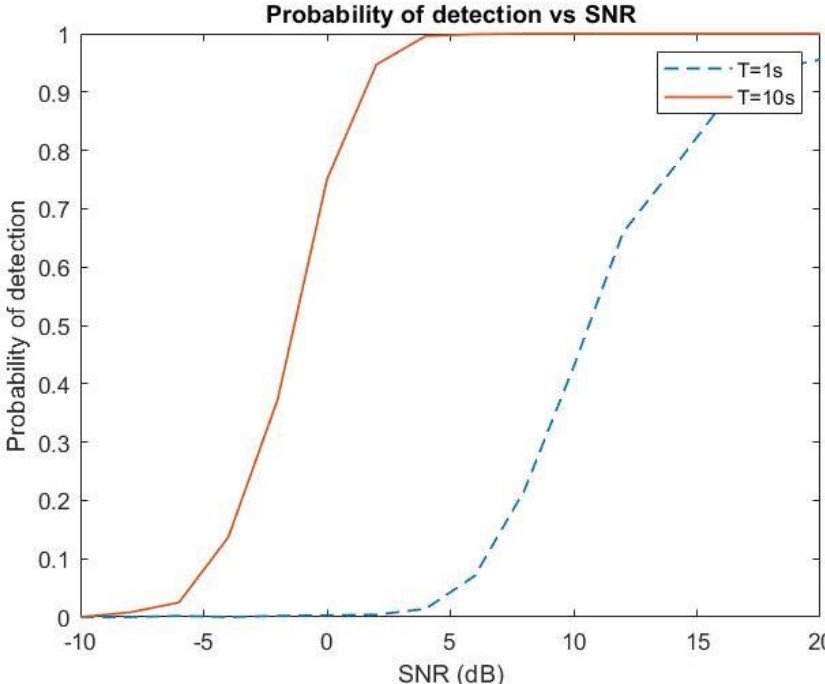

**Figure 6.** Detection probability curve with 10% False Alarm.

The performance of the detection test improves with increasing observation time $T$ as it was mentioned by [24,25]. Concerning the simulations of detection procedure, we used the Matlab implementation of CDP provided by Jerome Antoni [27,28]. All the technical aspects of CDP implementation, such as complexity, are presented in [27].

### 4.3. Estimation Algorithm

The theoretical model of signal previously introduced has several a priori properties that can be exploited: (1) the signal has line spectrum properties, i.e., it is sparse on the Fourier domain; (2) the micro-Doppler is periodic and cyclostationary, and thus, it can be decomposed in Fourier series; and (3) the fundamental $a_0$ has more energy than the harmonics as can be seen in Figures 3 and 4.

#### 4.3.1. Spectral Estimation Techniques

There are many spectral estimation techniques that can be used to estimate the different components of our signal, i.e., the central frequency $f_1$, the periodic frequency $f_0$, and the amplitudes of the line $(a_n)_{n \in \mathbb{Z}}$ [23,29]:

- Classical spectral estimation: these methods are called non-parametric methods because they do not require a priori information on the studied signal. One of the most famous methods is the periodogram based on the discrete Fourier transform.
- High-resolution methods: these methods are based on line spectrum (multiple tones) with additive white Gaussian noise. The main methods are: AR, Capon, and sub-space methods (MUSIC, ESPRIT.). The sub-space methods are particularly robust to noise.
- Sparse approximation: the sparsity consists of a signal to decompose it as a small number of atoms. These atoms can be part of a Fourier basis, wavelet basis, or other dictionaries (k-SVD, etc.). There exist many methods for estimating the different components such as basis pursuit, matching pursuit or thresholding techniques.
- Specific estimation techniques: it is possible to use an estimation technique that consists of the estimation of the parameters of a signal. For example, Niu et al. [30] proposed an estimation technique for sinusoidal frequency-modulated (SFM) signal. Note that SFM is a special case of our signal model where $f_1 = 0$ and $m(t) = A_v \sin(w_0 t)$.

#### 4.3.2. Proposed Algorithm

The signal's structure described in Equation (30) can be reformulated as follows:

$$X = D\alpha + N \qquad (40)$$

with:

- $X = [x(0) \dots x(L-1)]^T$: the vector containing the signal ($L \times 1$);
- $D = [V_{-N} \dots V_0 \dots V_N]$: the dictionary ($L \times (2N+1)$) containing the different tones $V_n$;
- $V_n = [1 \; e^{2\pi(f_1+nf_0)T_s} \; \dots \; e^{2\pi(f_1+nf_0)(L-1)T_s}]^T$: a vector containing the nth tone signal,
- $T_s$ the sampling time;
- $\alpha = [a_{-N} \dots a_0 \dots a_N]^T$: the amplitude vector containing the tones amplitude;
- $N = [n(0) \dots n(L-1)]^T$: the vector containing the additive white Gaussian complex noise.

The methods presented previously are either too general (spectral estimation methods) or too specific (parameter estimation methods). Here, we introduce the Parameters Estimation Algorithm Using Structured Dictionary (PEA-SD) technique that exploits all the a priori information known on the signal without being too limiting. It exploits: (1) the fundamental ($f_1$), which has an amplitude greater than the harmonics; (2) the micro-Doppler term, which is periodic ($f_0$); (3) the line spectrum has a specific structure ($f_1 + nf_0$ with $n \in \mathbb{Z}$). One important step of the algorithm exploits the structure of the signal using a dictionary to estimate the amplitudes. This type of approach has been developed by Mototolea et al. [31] for drone detection purposes using a priori information on the signal

structure. Furthermore, the maximum likelihood technique used for amplitudes estimation in high-resolution methods (MUSIC, ESPRIT, etc.) is also based on this type of structured dictionary as explained by Badeau in [32]. Our proposal is the following (see Algorithm 1):

---
**Algorithm 1:** Proposed algorithm

---
**Result:** The periodic frequency $\hat{f}_0$, the central frequency $\hat{f}_1$ and the amplitudes $\hat{\alpha}$

Estimate $f_1$;

Estimate $f_0$;

Construct the dictionary $D$;

Estimate $\alpha$.

---

For each parameter's estimation, it is possible to use several methods:

- Estimate $f_1$: Maximum likelihood method [26], high-resolution methods (MUSIC, ESPRIT), etc.
- Estimate $f_0$: Autocorrelation function (AF), Cyclic Autocorrelation Function [30], etc.
- Estimate $\alpha$:
  - Maximum likelihood [32]: $\hat{\alpha} = (D^H D)^{-1} D^H X$
  - Maximum a posteriori (Derived from the associated loss function: $\|D\alpha - X\|^2 + \gamma \sum_{n=-N}^{N} c_n a_n^2 = (D\alpha - X)^H (D\alpha - X) + (\gamma \alpha^H C\alpha)$: $\hat{\alpha} = (D^H D + \gamma C)^{-1} D^H X$.

As previously mentioned, in the last step of the proposed algorithm, it is possible to use a maximum a posteriori estimation procedure to estimate the amplitudes $\alpha$ using supplementary a priori knowledge. Indeed, the fundamental $a_0$ is usually greater than the harmonics amplitude $(a_n)_{n \in [\![-N;N]\!]^*}$ (see Figure 3). It is then possible to use a constraint matrix $C$ to impose the constraints during the amplitudes estimation. The matrix $C = diag([c_{-N} \ldots c_0 \ldots c_N]^T)$ is diagonal and is equivalent to the following constraint $\sum_{n=-N}^{N} c_n a_n^2$. This constraint method is a generalization of the ridge regularization $\sum_{n=-N}^{N} a_n^2$ (a case where C is an identity matrix), which is widely used in machine learning to reduce the parameter amplitudes of a linear model. From a statistical point of view, enforcing the constraint matrix $C$ to estimate the amplitude is equivalent to modeling the amplitude $\alpha$ as a random variable depending on a multivariate normal complex distribution with mean vector $\mu = [0 \ldots 0]^T$ and a covariance matrix $\Sigma = (\gamma C)^{-1}$. Note that the parameter $\gamma$ is a hyperparameter that makes a trade-off between reconstruction error $\|D\hat{\alpha} - X\|^2$ and amplitude constraints $\sum_{n=-N}^{N} c_n a_n^2$.

It is also possible to estimate the noise variance $\sigma^2$ as follows [32]:

$$\hat{\sigma^2} = \frac{1}{L} \|D\hat{\alpha} - X\|^2 \tag{41}$$

Note that the proposed algorithm corresponds to a family of algorithms. Indeed, each step (except dictionary construction) is not dependent on a particular method. For example, the estimation of $f_1$ can be done using either Fourier transformation or high-resolution methods such as MUSIC. This allows for the creation of an algorithm highly adapted for a particular situation.

### 4.3.3. Results

As previously mentioned, one hundred simulations were realized to compare performances over different observations times $T$ and different SNR values with sampling frequency $F_s = 1000$ Hz. The signal used for the simulations is, as previously explained, a sinusoidal frequency-modulated (SFM) signal with a frequency offset component and additive white Gaussian complex noise:

$$\tilde{s}(t) = e^{j\beta \sin(2\pi f_0 t)} e^{j2\pi f_1 t} + \tilde{n}(t) \tag{42}$$

with:

- $\beta = 0.1$: the modulation index;
- $f_0 \in [30, 150]$: the periodic frequency;
- $f_1 \in [-50, 50]$: the frequency offset;
- $\tilde{n}(t)$: the additive white Gaussian complex noise.

In this section, we compare several estimation methods corresponding with our signal model:

- Matching pursuit: a sparse approximation algorithm using a greedy approach [33]. The dictionary used for sparse approximation is a Fourier redundant dictionary with frequency spacing of 0.1 Hz.
- Root-MUSIC: a high-resolution algorithm using noise subspace properties [32]. We use the "rootmusic" Matlab function to estimate the different frequency components.
- Proposed algorithm: the estimation algorithm we proposed that exploits all the a priori information derived from the signal model using Equations (43) and (44). The frequency precision used to estimate $f_1$ is 0.01 Hz.

The estimation of $f_1$ is performed using [26]:

$$\hat{f}_1 = arg\ max_f\ |\tilde{Y}(f)|^2 \tag{43}$$

The estimation of $f_0$ is performed using [24]:

$$\hat{f}_0 = arg\ max_{\alpha \in [\alpha_{min}, \alpha_{max}]}\ I(\alpha) \tag{44}$$

The metric used to compared the different methods is mean square reconstruction error (MSRE) (also called reconstruction MSE):

$$MSRE = \frac{\|\tilde{S} - D\hat{\alpha}\|^2}{L} \tag{45}$$

with:

- $L$: the signal length;
- $\tilde{S} = [\tilde{s}(0), \ldots, \tilde{s}(L-1)]^T$: the signal $\tilde{s}(t)$;
- $D$: the dictionary used for estimation (depending of the method);
- $\hat{\alpha}$: the estimated weight.

Our proposed algorithm gives better results than matching pursuit or Root-MUSIC for different time durations $T$ and even for low SNR values (see Figure 7a,b). One can argue that the better performance of our method over matching pursuit is due to the relative higher precision of the Fourier transform. However, we tried to build a dictionary using a frequency precision of 0.01 Hz, but the memory required was too high to perform matching pursuit because for $T = 10$ s the dictionary size is $(10^5 * 10^4)$. Note that all studied estimation techniques estimate the atoms/components of a dictionary then perform amplitude estimation. So, the amplitude estimation method used for all estimation methods is the maximum likelihood.

### 4.4. Hybrid Detection/Estimation Procedure

As we can see, the statistical detection test and the estimation procedure require the same tools. Indeed, the estimation of $f_1$ is based on the Fourier transform, which is computed at the first step of a detection test to synchronize the signal and the hypothesis test detects the cyclic frequency $f_0$ present on the signal using a cyclic frequency domain profile (CDP). So, for practical uses, it can be beneficial to use a hybrid detection/estimation procedure, as described in Figure 8.

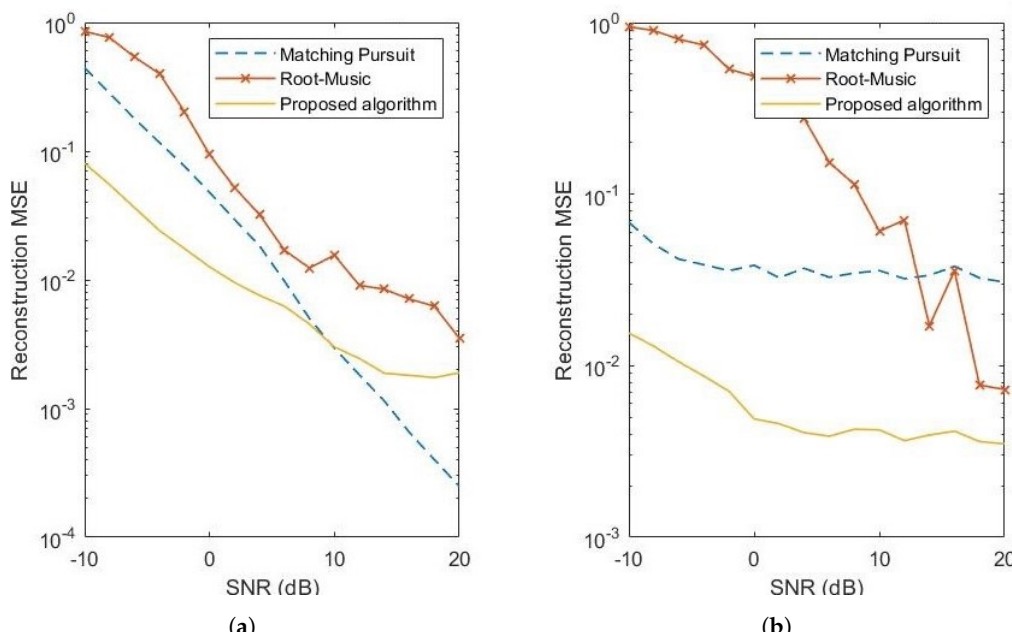

(**a**)  (**b**)

**Figure 7.** (**a**) Mean square reconstruction error against SNR for $T = 1$ s. (**b**) Mean square reconstruction error against SNR for $T = 10$ s.

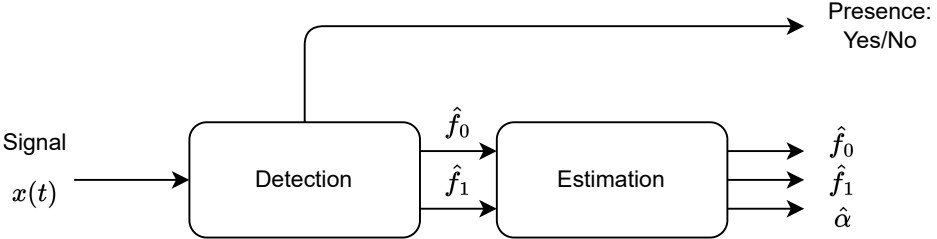

**Figure 8.** Hybrid detection/estimation procedure.

## 5. Future Use Cases

In this section, we present future use cases exploiting the Doppler modulation effect especially for micro-Doppler in a radio communications context.

The previous sections were widely theoretical, trying to formalize the physical phenomenon and the signal processing problems (signal cancellation, detection, and estimation) allowing exploitation of this phenomenon. The following subsection presents several use cases of the Doppler modulation specifically micro-Doppler modulation, i.e., $x(t) = m(t)$, a micro-movement such as vibration. Four different use cases are presented in this section.

### 5.1. Simulation

The Doppler modulation effect can be used to simulate multipath channel configurations under complex movement. The presented geometrical model is restricted to a unique direct path, but as previously mentioned, it is possible to model a multipath configuration using virtual sources [5]. This modeling approach can be another perspective different from classical models such as multipath under differential Doppler [2] or statistical channel models [3] based on Clarke model [4]. Note that a common characteristic of these models, including ours, is that all are based on ray tracing theory.

### 5.2. Vibration Object Detection

The micro-Doppler can be used for detection of vibrating objects such as drones. Using the proposed binary hypothesis test, it is possible to detect vibrating objects. Furthermore, our estimation algorithm can be used to estimate the properties of the signal, i.e., the vibration frequency $f_0$ and the amplitude $\alpha$. For example, as mentioned in Matthan [13],

the vibration frequency $f_0$ can be used to estimate the size of a drone. The amplitude $\alpha$ can be used to differentiate drones of the same model or to classify specific movements (hovering, etc.).

### 5.3. RF Fingerprinting

RF fingerprinting, or radio fingerprinting, is a set of techniques allowing for the identification of a transmitter by the imperfection of its components. These techniques are used to secure communications as a non-cryptographic authentication method or to detect intrusions/wireless attacks. The estimated parameters $\hat{f}_0$ and $\hat{\alpha}$ can be used as a set of new features for RF fingerprinting method. Furthermore, these features can be associated with classical features for RF fingerprinting approaches, such as those proposed in [34] (I/Q offset, etc.).

### 5.4. Vibration Analysis

The micro-Doppler effect can be used for vibration analysis allowing, for example, the detection of mechanical gear bearing vibration or similar default problems [21]. To perform this analysis, a transmitter is fixed on a monitored machine and the micro-Doppler parameters are extracted from the communication at the receiver side. This approach would allow for a cheaper and simpler monitoring method than other classical methods based on sensors.

## 6. Conclusions

The Doppler modulation effect corresponds to the effect of complex movement on received signal. Particularly, in this paper, we focused on the micro-Doppler effect produced by vibration. To the best of our knowledge, this study is the first attempt to model it for radio communication systems. Moreover, we also have proposed signal processing techniques to exploit this phenomenon among communication signal cancellation, detection, and estimation. The communication signal cancellation step allows for the removal of communication signal (M-PSK) from the received signal to facilitate analysis of Doppler modulation. Then, the detection step allows for the detection of the presence of micro-Doppler in the communication. Finally, we proposed an estimation algorithm, called PEA-SD, designed to estimate micro-Doppler parameters, and it was demonstrated that it outperforms classic spectral estimation algorithms. These tools would allow for the development of RF sensing applications based on communication itself at the physical layer. Finally, this work can also make some connections with other domains such as the micro-Doppler effect in radar, especially in passive radar.

Concerning further works, we intend to explore new signal processing techniques such as signal cancellation techniques for other modulations. Moreover, the exploration of new methods to make micro-Doppler effect detection and estimation less sensitive to observation time is also another important topic. Finally, developing use cases is paramount to prove that the Doppler modulation effect in radio communications is not just a theoretical problem but can afford real-world applications.

**Author Contributions:** Conceptualization, L.M.-R.; Formal analysis, L.M.-R.; Funding acquisition, R.G.; Investigation, L.M.-R. and C.C.; Methodology, L.M.-R., D.L.J., F.L.R., C.C. and R.G.; Writing—original draft, L.M.-R.; Writing—review & editing, F.L.R., C.C. and R.G. All authors have read and agreed to the published version of the manuscript.

**Funding:** This work was funded by ENSTA Bretagne of Brest and supported by the IBNM (Brest Institute of Computer Science and Mathematics) CyberIoT Chair of Excellence of the University of Brest. This work has been developed for the program "AN DRO" (Analyse Numérique de signaux de Drones).

**Data Availability Statement:** Not applicable.

**Acknowledgments:** The authors acknowledge ENSTA Bretagne of Brest and the IBNM CyberIoT Chair of Excellence of the University of Brest for their supports. The authors also acknowledge Morgan Fassier, research engineer, for their implication in this work.

**Conflicts of Interest:** The authors declare no conflict of interest.

## Appendix A. Doppler Modulation: Geometrical Modeling Simplifications

The goal of this appendix is to formulate the received signal dependence on transmitter movement $\vec{x}(t)$ using the previously introduced relation $r(t) = A(t - \tau_i(t))s(t - \tau_i(t))$. To do so, we proceed in three steps to obtain Equation (7).

### Appendix A.1. Step 1: Geometrical Simplification

The vector $\vec{l}_i(t) = \vec{l}_i(0) - \vec{x}(t)$ is the distance between the transmitter and the receiver and depends on time. Firstly, to simplify $\|\vec{l}_i(t)\|$, the expression of $\|\vec{l}_i(t)\|^2$ needs to be simplified:

$$\|\vec{l}_i(t)\|^2 = \|\vec{l}_i(0) - \vec{x}(t)\|^2 \tag{A1}$$

$$= \|\vec{l}_i(0)\|^2 - 2\langle\vec{x}(t), \vec{l}_i(0)\rangle + \|\vec{x}(t)\|^2 \tag{A2}$$

$$= \|\vec{l}_i(0)\|^2 - 2\|\vec{l}_i(0)\|\langle\vec{x}(t), \vec{l}_{i0}\rangle + \|\vec{x}(t)\|^2 \tag{A3}$$

where $\vec{l}_i(0) = \|\vec{l}_i(0)\|\vec{l}_{i0}$.

Considering hypothesis 1 ($\|\vec{x}(t)\| \ll \|\vec{l}_i(0)\|$), the following approximation can be done:

$$\|\vec{l}_i(t)\|^2 \approx \|\vec{l}_i(0)\|^2(1 - 2\frac{\langle\vec{x}(t), \vec{l}_{i0}\rangle}{\|\vec{l}_i(0)\|}) \tag{A4}$$

$\|\vec{l}_i(t)\|$'s dependence on $\|\vec{l}_i(t)\|^2$ can then be determined using a first-order Taylor approximation of the square root function and hypothesis 1:

$$\|\vec{l}_i(t)\| = \sqrt{\|\vec{l}_i(t)\|^2} \tag{A5}$$

$$\approx \sqrt{\|\vec{l}_i(0)\|^2(1 - 2\frac{\langle\vec{x}(t), \vec{l}_{i0}\rangle}{\|\vec{l}_i(0)\|})} \tag{A6}$$

$$\approx \|\vec{l}_i(0)\|(1 - \frac{\langle\vec{x}(t), \vec{l}_{i0}\rangle}{\|\vec{l}_i(0)\|}) \tag{A7}$$

$$\approx \|\vec{l}_i(0)\| - \langle\vec{x}(t), \vec{l}_{i0}\rangle \tag{A8}$$

### Appendix A.2. Step 2: Signal Simplification

The Friis equation is used to expressed received power $P_r$ for direct path propagation:

$$P_r(t) = P_t G_t G_r(\frac{\lambda}{4\pi\|\vec{l}_i(t)\|})^2 \tag{A9}$$

with:

- $P_r(t)$: the received power;
- $P_t$: the transmitted power;
- $G_t$: the transmitter antenna gain;
- $G_r$: the receiver antenna gain.

So, the propagation loss term $A(t)$ becomes:

$$A(t) = \sqrt{P_r(t)} \tag{A10}$$

$$= \sqrt{P_t G_t G_r (\frac{\lambda}{4\pi \|\vec{l_i}(t)\|})^2} \tag{A11}$$

$$= \sqrt{P_t G_t G_r (\frac{\lambda}{4\pi})^2 \frac{1}{\|\vec{l_i}(t)\|}} \tag{A12}$$

Using hypothesis 2 ($\left\|\vec{l_i}(t)\right\| \approx \left\|\vec{l_i}(t - \tau_i(t))\right\|$) and Equation (A12), the received signal will be:

$$r(t) = A(t - \tau_i(t))s(t - \tau_i(t)) \tag{A13}$$

$$\approx A(t)s(t - \tau_i(t)) \tag{A14}$$

Note that received signal formulation is equivalent to the formulation presented in [1] in direct path propagation configuration.

Using the result from step 1 ($\|\vec{l_i}(t)\| \approx \|\vec{l_i}(0)\| - \langle \vec{x}(t), \vec{l_{i0}} \rangle$), the received signal becomes:

$$r(t) = A(t)s(t - \frac{\|\vec{l_i}(0)\|}{c} + \frac{\langle \vec{x}(t), \vec{l_{i0}} \rangle}{c}) \tag{A15}$$

$$= A(t)s(t - \tau_0 + \frac{\langle \vec{x}(t), \vec{l_{i0}} \rangle}{c}) \tag{A16}$$

with $\tau_0 = \frac{\|\vec{l_i}(0)\|}{c}$ as the initial delay.

$A(t)$ can also be approximate as follows:

$$A(t) = \sqrt{P_t G_t G_r (\frac{\lambda}{4\pi})^2 \frac{1}{\|\vec{l_i}(t)\|}} \tag{A17}$$

$$\approx \sqrt{P_t G_t G_r (\frac{\lambda}{4\pi})^2 \frac{1}{\|\vec{l_i}(0)\| - \langle \vec{x}(t), \vec{l_{i0}} \rangle}} \tag{A18}$$

$$\approx \sqrt{P_t G_t G_r (\frac{\lambda}{4\pi \|\vec{l_i}(0)\|})^2 \frac{1}{1 - \frac{\langle \vec{x}(t), \vec{l_{i0}} \rangle}{\|\vec{l_i}(0)\|}}} \tag{A19}$$

$$\approx A \frac{1}{1 - \frac{\langle \vec{x}(t), \vec{l_{i0}} \rangle}{\|\vec{l_i}(0)\|}} \tag{A20}$$

with $A = \sqrt{P_t G_t G_r (\frac{\lambda}{4\pi \|\vec{l_i}(0)\|})^2}$.

Furthermore, we obtain the following received signal formulation:

$$r(t) = A \frac{1}{1 - \frac{\langle \vec{x}(t), \vec{l_{i0}} \rangle}{\|\vec{l_i}(0)\|}} a(t - \tau_0 + \frac{\langle \vec{x}(t), \vec{l_{i0}} \rangle}{c})$$

$$\cos(2\pi f_c t + k\langle \vec{x}(t), \vec{l_{i0}} \rangle + \phi(t - \tau_0 + \frac{\langle \vec{x}(t), \vec{l_{i0}} \rangle}{c}) - \phi_0) \tag{A21}$$

with $\phi_0 = 2\pi f_c \tau_0$ as the initial phase.

Furthermore, its corresponding analytic signal is:

$$r_a(t) = A \frac{1}{1 - \frac{\langle \vec{x}(t), \vec{l_{i0}} \rangle}{\|\vec{l_i}(0)\|}} \tilde{s}(t - \tau_0 + \frac{\langle \vec{x}(t), \vec{l_{i0}} \rangle}{c}) e^{-j\phi_0} e^{jk\langle \vec{x}(t), \vec{l_{i0}} \rangle} e^{j2\pi f_c t} \tag{A22}$$

*Appendix A.3. Step 3: Neglecting the Amplitude Modulation Due to Motion*

The goal of this step of the demonstration is to neglect the amplitude modulation due to motion. Using the analytic signal, we can observe two forms of modulation due to movement $\vec{x}(t)$:

- $\dfrac{1}{1 - \frac{\langle \vec{x}(t), \vec{l_{i0}} \rangle}{\|\vec{l_i}(0)\|}}$: an amplitude modulation

- $e^{jk\langle \vec{x}(t), \vec{l_{i0}} \rangle}$: a phase modulation.

In two different ways, depending on type of movement $\vec{x}(t)$, the amplitude modulation can be neglected in front of the received signal. On the one hand, for an arbitrary movement $\vec{x}(t)$, using hypothesis 3a, the amplitude $A(t)$ is considered as constant: $A(t) \approx A$. This hypothesis is more general than hypothesis 3b and can be considered to be a stronger hypothesis. One the other hand, for a small movement $\vec{x}(t)$ ($k\|\vec{x}(t)\| \ll 1$), the amplitude modulation can be neglectable compared to phase modulation. To do so, we can compare the first-order Taylor series of both terms:

- $\dfrac{1}{1 - \frac{\langle \vec{x}(t), \vec{l_{i0}} \rangle}{\|\vec{l_i}(0)\|}} \approx 1 + \dfrac{\langle \vec{x}(t), \vec{l_{i0}} \rangle}{\|\vec{l_i}(0)\|}$

- $e^{jk\langle \vec{x}(t), \vec{l_{i0}} \rangle} \approx 1 + j\langle \vec{x}(t), \vec{l_{i0}} \rangle$.

Considering hypothesis 3b ($k\|\vec{l_i}(0)\| \gg 1$), the term $\dfrac{\langle \vec{x}(t), \vec{l_{i0}} \rangle}{\|\vec{l_i}(0)\|}$ is negligible in terms of magnitude compared to $j\langle \vec{x}(t), \vec{l_{i0}} \rangle$. Thus, the amplitude modulation can be neglected in front of the phase modulation (micro-Doppler): $A(t) \approx A$. This hypothesis is considered as less general because it is restricted to small motions. It can be noted that $\beta$ mentioned in signal used for simulations described in Equations (39) and (42) is equal to $max_t k\|\vec{x}(t)\|$.

Finally, considering all these analytical developments, the received signal can be formulated as follows:

$$r(t) = A a\left(t - \tau_0 + \frac{\langle \vec{x}(t), \vec{l_{i0}} \rangle}{c}\right)$$

$$\cos\left(2\pi f_c t + k\langle \vec{x}(t), \vec{l_{i0}} \rangle + \phi\left(t - \tau_0 + \frac{\langle \vec{x}(t), \vec{l_{i0}} \rangle}{c}\right) - \phi_0\right) \quad \text{(A23)}$$

## Appendix B. Noise Modeling for Signal Cancellation Method

In this appendix, we demonstrate the following approximation resulting from signal cancellation method (Equation (24)):

$$\tilde{r}(t) = (A\tilde{s}(t)e^{jkx(t)} + \tilde{n}(t))^M \quad \text{(A24)}$$

$$\approx A^M e^{jkMx(t)} + MA^{M-1}\tilde{s}(t)^{M-1}\tilde{n}(t)e^{jk(M-1)x(t)} \quad \text{(A25)}$$

$$\approx A^M e^{jkMx(t)} + \tilde{n}_1(t) \quad \text{(A26)}$$

The demonstration will be divided in two parts:

- Proof showing that the term $MA^{M-1}\tilde{s}(t)^{M-1}\tilde{n}(t)e^{jk(M-1)x(t)}$ is a white Gaussian complex noise;
- Proof showing that the term $\sum_{m=2}^{M} \tilde{s}(t)^k e^{jkmx(t)}\tilde{n}(t)^{M-k}$ is negligible compared to $\tilde{n}_1(t)$.

Note that $\tilde{s}'(t) = \tilde{s}(t)^{M-1}$ is a cyclostationary signal corresponding to the complex conjugate of $\tilde{s}(t)$ with the following properties: $\mathbb{E}(\tilde{s}'(t)) = 0$, $\mathbb{E}(\tilde{s}'(t)\tilde{s}'(t)^*) = 1$, and $\mathbb{E}(\tilde{s}'(t)\tilde{s}'(t-\tau)^*) = \sum_{k=-\infty}^{+\infty} \pi(\frac{t-kT_0}{T_0})^* \pi(\frac{t-\tau-kT_0}{T_0})$.

*Appendix B.1. Properties of the First Term of the Noise*

The term $\tilde{n}_1(t)$ is the product between different terms: a Gaussian noise $\tilde{n}(t)$, a cyclostationary signal $\tilde{s}'(t)$, a deterministic signal $e^{jk(M-1)x(t)}$, and a factor $A^{M-1}M$.

The expected value of $\tilde{n}_1(t)$ is:

$$\mathbb{E}(\tilde{n}_1(t)) = 0 \tag{A27}$$

The instantaneous energy of $\tilde{n}_1(t)$ is:

$$\mathbb{E}(\tilde{n}_1(t)\tilde{n}_1(t)^*) = |MA^{M-1}|^2 \tag{A28}$$

The autocorrelation of $\tilde{n}_1(t)$ is:

$$\mathbb{E}(\tilde{n}_1(t)\tilde{n}_1(t-\tau)^*) = |MA^{M-1}|^2\delta(\tau) \tag{A29}$$

The resulting noise expression can be considered as wide sense stationary process.

To prove that the resulting noise has Gaussian properties, we study their modulus and their phase independently. Note that a Gaussian complex white noise $\tilde{n}(t) \sim \mathcal{CN}(0,1)$ implies the following properties: the modulus depends on a Rayleigh distribution ($|\tilde{n}(t)| \sim Rayleigh(1/\sqrt{2})$) and phase depends on a uniform distribution ($arg(\tilde{n}(t)) \sim U([0, 2\pi])$).

The modulus of the resulting noise is:

$$|\tilde{n}_1(t)| = |MA^{M-1}||\tilde{n}(t)| \tag{A30}$$

Note that the factor $MA^{M-1}$ has no impact on this demonstration because it influences the modulus, and the resulting modulus is a random variable following a Rayleigh distribution.

Demonstration: we will consider $Y = cX$ with a constant $c > 0$ and $X$ a random variable following a Rayleigh distribution with a scale parameter $\sigma$, a probability density function $f$, and a cumulative distribution function $F$.

$$P(Y < y) = P(cX < y) \tag{A31}$$

$$= P(X < \frac{y}{c}) \tag{A32}$$

$$= F(\frac{y}{c}) \tag{A33}$$

To compute the resulting probability density function $g$, we derive the term:

$$g(y) = \frac{1}{c}f(\frac{y}{c}) \tag{A34}$$

$$= \frac{1}{c}\frac{y/c}{\sigma^2}e^{(-\frac{(y/c)^2}{2\sigma^2})} \tag{A35}$$

$$= \frac{y}{(c\sigma)^2}e^{(-\frac{y^2}{2(c\sigma)^2})} \tag{A36}$$

Now, the resulting random variable Y follows a Rayleigh distribution with the scale parameter $\sigma' = c\sigma$ (here $c = |MA^{M-1}|$).

The phase of the resulting noise is:

$$arg(\tilde{n}_1(t)) = arg(\tilde{n}(t)) + arg(\tilde{s}(t)) + k(M-1)x(t) \tag{A37}$$

The modulus of the resulting noise follows a Rayleigh distribution, and the phase follows a uniform distribution between 0 and $2\pi$. Indeed, $arg(\tilde{s}(t)$ is stationary in part which means $arg(\tilde{n}_1(t)) \sim U([arg(\tilde{s}(t)) + k(M-1)x(t), 2\pi + arg(\tilde{s}(t)) + k(M-1)x(t)])$; however, due to modulo property of the complex exponential, is similar to $arg(\tilde{n}_1(t)) \sim U([0, 2\pi])$.

We have proven that the resulting noise is wide-sense stationary and has Gaussian complex properties. So, we can consider the resulting noise $\tilde{n}_1(t)$ as a white Gaussian complex noise.

*Appendix B.2. Neglecting the Second Part of the Noise*

This proof will be performed in two steps: (1) comparison of the noise $\tilde{n}_1(t)$ and the term $\tilde{n}_2(t) = A^{M-2}\frac{M(M-1)}{2}\tilde{s}(t)^{M-2}e^{jk(M-2)x(t)}\tilde{n}(t)^2$, which that can be considered as the predominant term (from a power point-of-view); (2) determination of the threshold, which depends of the SNR, to allow us to neglect the second part of the noise.

We calculate the expected value of the instantaneous amplitude for $\tilde{n}_1(t)$:

$$\mathbb{E}(|\tilde{n}_1(t)|) = A^{M-1}M\mathbb{E}(|\tilde{n}(t)|) \tag{A38}$$

$$= A^{M-1}M\sqrt{\frac{\pi}{4}} \tag{A39}$$

Furthermore, we calculate the expected value of the instantaneous amplitude for $\tilde{n}_2(t)$:

$$\mathbb{E}(|\tilde{n}_2(t)|) = A^{M-2}\frac{M(M-1)}{2}\mathbb{E}(|\tilde{n}(t)|^2) \tag{A40}$$

$$= A^{M-2}M(M-1) \tag{A41}$$

Note that $|\tilde{n}(t)| \sim Rayleigh(1/\sqrt{2})$ and $|\tilde{n}(t)|^2 \sim Exponential(1)$.
The ratio between the two term is the following:

$$\frac{\mathbb{E}(|\tilde{n}_1(t)|)}{\mathbb{E}(|\tilde{n}_2(t)|)} = \frac{A\sqrt{\pi/4}}{M-1} \tag{A42}$$

The associated power ratio is the following:

$$\frac{\mathbb{E}(|\tilde{n}_1(t)|)^2}{\mathbb{E}(|\tilde{n}_2(t)|)^2} = \frac{A^2\pi}{4(M-1)^2} \tag{A43}$$

It is possible to define a specific threshold $C_n$ which allows us to neglect $\tilde{n}_2(t)$ in front of $\tilde{n}_1(t)$:

$$A^2 > \frac{4}{\pi}(M-1)^2 C_n \tag{A44}$$

The SNR[dB], i.e., signal-to-noise ratio (SNR) for the signal presented in Equation (16), corresponds to $10log_{10}(A^2)$ because the noise $\tilde{n}(t)$ has unitary power and similarly for $\tilde{s}(t)$ and $e^{jkx(t)}$. For example, if we consider a BPSK with $C_n = 10$, the corresponding SNR[dB] is approximately 11 dB. If we omit the second term of the noise after the communication signal cancellation, the resulting SNR corresponds to $10\log_{10}(\frac{A^4}{4A^2}) = $ SNR[dB] $- 6$ dB. The power of each harmonic also increases because the Doppler modulation term is powered by M. For example, in the case of an SFM ($\tilde{s}(t) = A\tilde{s}(t)e^{j\beta\sin(w_0t} + \tilde{n}(t))$), the power of first-order harmonic will increase by 6 dB because $J_1(2\beta) \approx 2J_1(\beta)$.

## Appendix C. Modified Binary Hypothesis Test

This appendix will explain the modified version of the binary hypothesis test used for micro-Doppler detection.

*Appendix C.1. Initial Binary Hypothesis Test*

Our detection method is based on the work of Kim et al. [24] for detection of modulation under additive white Gaussian noise. In [25], Chen et al. extended this statistical test to cyclostationary signals. This test is based on the so-called cyclic frequency domain profile (CDP) (Equation (33)):

$$I(\alpha) = max_f|C_x^\alpha(f)| \tag{A45}$$

with:

- $C_x^\alpha(f)$ the spectral coherence,

- $\alpha$ the cyclic frequency (here equivalent to $f_0$).

The hypotheses required by their method are:

$$\begin{aligned} H_0 &: n(t) \\ H_1 &: s(t) + n(t) \end{aligned} \tag{A46}$$

with:

- $s(t)$: A modulation among DSB-SC AM, BPSK, b-FSK, MSK and QPSK (or cyclostationary signals [25]);
- $n(t)$: An additive white Gaussian complex noise (AWGN).

The authors proposed a binary hypothesis test based on the following statistic:

$$C_I = \frac{I(\alpha)}{\sqrt{\frac{1}{N}\sum_{\alpha=0}^{N} I(\alpha)^2}} \tag{A47}$$

Furthermore, they proposed a specific threshold computed for the null hypothesis:

$$C_{TH} = \frac{max(I(\alpha))}{\sqrt{\frac{1}{N}\sum_{\alpha=0}^{N} I(\alpha)^2}} \tag{A48}$$

Binary hypothesis testing is performed as follows:

$$C_I \leq C_{TH} : Declare\ H_0 \tag{A49}$$

$$C_I > C_{TH} : Declare\ H_1 \tag{A50}$$

In [24], the output of the hypothesis testing procedure was a vector where all the CDP values greater than $C_{TH}$ are encoded as one and the others are null. The authors used this vector for classification purposes in a cognitive radio context. In our case, the output vector corresponds to zero for all the values lower or equal to $C_{TH}$ and $C_I$ for the values greater than $C_{TH}$. For hybrid detection estimation procedure, the periodic frequency $f_0$ is estimated by taking the argument corresponding to the maximum value of this vector.

*Appendix C.2. Modified Hypothesis Test*

The initial hypotheses used are presented in Equation (32) and the resulting hypotheses obtained after transformation step $h(\tilde{y})$ (Equation (34)) are presented in Equation (35).

In our case, we know that the cyclic frequency is contained between a certain interval $[\alpha_{min}, \alpha_{max}]$. For example, for drone detection purposes, Nguyen et al. [13] shows that the vibration frequencies of drones are between 35 and 140 Hz. So, using expert domain knowledge, it is possible to restrict the research space. In the simulations, we restrict the research space $\alpha \in [30, 150]$.

We know that the null hypothesis does not have cyclic frequency in $[\alpha_{min}, \alpha_{max}]$, even if there is still a frequency offset component due to frequency estimation error.

We set $C_{TH}$ using a Neyman–Pearson procedure. First of all, we estimate the probability density function (PDF) and the cumulative distribution function (CDF) by a frequentist approach. It consists of the simulation of many realizations (here, one thousand) to estimate the PDF and CDF based on the law of large numbers. Figure A1a,b correspond, respectively, to the empirical CDF for $T = 1$ s and $T = 10$ s. Note that we take the worst-case SNR = $-10$ dB that gives the lowest $C_{TH}$ values compared to a higher SNR. Using the estimated cumulative density function, we find the threshold to obtain the specific false alarm rate (here 10%).

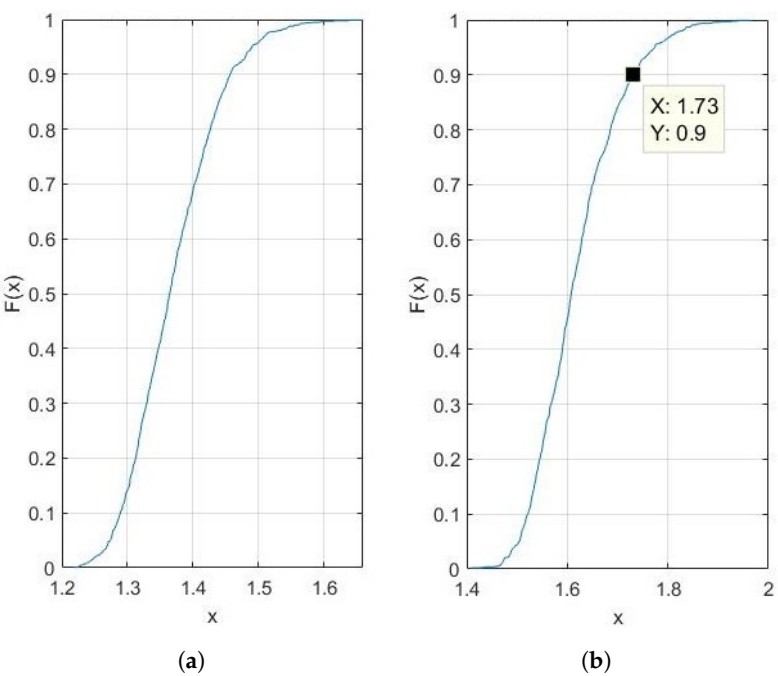

(**a**)                                                                                         (**b**)

**Figure A1.** (**a**) Empirical CDF for *T* = 1 s. (**b**) Empirical CDF for *T* = 10 s.

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
