# Peer review of "From Modeling to Sensing of Micro-Doppler in Radio Communications"

_remotesensing, doi:10.3390/rs14246310_

Round 1

Reviewer 1 Report

·           Some grammatical and a few spelling errors are present, moderate English proof reading and corrections are required.

·        For better clarity, the authors should describe how formula 23 is derived from formula 22.

·        A model describing the signal modulation due to micro-movement is provided with formula 23. However, there is no real comparison performed with the measurement results in Figure 3. The evidence for the applicability of the formula for the conducted experiment needs to be presented.

·         It is mentioned in footnote 2 that measurements with different parameters (voltage, frequency and distance) were performed, but the results are not shown in the publication. It would be interesting for the reader to know if and where these results will be published. Otherwise the provided content appears incomplete.

·        More descriptions and explanation for the results in Figure 3 should be given. The relation between the results and the theoretical signal model in formula 23 needs to be verified and some additional points should be addressed:

·        What exact frequencies were expected in Section 2.3? What exact frequencies were measured?

·        With 6000 rpm about 100Hz of vibrational frequency could be expected. Why is there a discrepancy?

·        The central peak in Figure3a is not explained. It can be assumed that it is the direct-signal. However, a frequency offset of several Hertz is present.

·        The labeling of Figure 3 (a) is cut off.

·        The abbreviation MPSK should be explained.

·        Footnote 3 indicates that further experiments with BPSK, QPSK were performed. The results would be of interest for this journal, but they are not shown. Thus, the presented results appear incomplete. It would be interesting for the reader to know if and where these results will be published.

·        Labels (a) and (b) in Figure 4 are partially cut off.

·        Section 4.2 gives a theoretical explanation of a method for micro-movement detection is given.  Afterwards simulation results are provided. Why wasn’t the method applied to the measured data from the previous sections?

·        Please provide references to the list of estimation techniques in section 4.3.1.

·        The authors should provide more explanation behind the “Reconstruction MSE” value in Figure 7.

·        The graphs in Figure 7 are partially cut off.

·        The conclusions mention the connection to micro-Doppler in radar, especially in passive radar. A relevant study on the measurement of vibrations in wind-turbines using passive radar is presented in the following paper. It also gives a possible real-world application for micro-Doppler estimation in radio communications.

Author Response

Some grammatical and a few spelling errors are present, moderate English proof reading and corrections are required.
For better clarity, the authors should describe how formula 23 is derived from formula 22.
The formula 23 come from Jacobi-Anger expansion, we added in parenthesis in section 2.3.1.
A model describing the signal modulation due to micro-movement is provided with formula 23. However, there
is no real comparison performed with the measurement results in Figure 3. The evidence for the applicability of the formula for the conducted experiment needs to be presented.
→ The paragraph 2.3.2 has been updated to better explained the link between the theoretical model and the measurements.
It is mentioned in footnote 2 that measurements with different parameters (voltage, frequency and distance) were performed, but the results are not shown in the publication. It would be interesting for the reader to know if and where these results will be published. Otherwise the provided content appears incomplete.
This footnote has been removed due to the fact that the mentioned experiments were probably part of an extended version of this article. We provide to reviewer 1 a document describing the mentionned experiments. Specifically, the experiments about the influence of the distance, the vibrator voltage supply and the frequency are described in section 1 of this document.
More descriptions and explanation for the results in Figure 3 should be given. The relation between the results and the theoretical signal model in formula 23 needs to be verified and some additional points should be addressed:
We modified figure 3.a to correct the frequency offset due to the fact the transmitter and the receiver are not synchronised (neither in frequency or in time). One major results of this correction is the frequency peaks from IMU (recording vibration from vibrator) and the frequency peaks from
baseband signal (recording from the receiver) are now synchronised. Thus, showing the spectral density of the received signal depend on vibration from vibrator as predicted by theoretical model.
- What exact frequencies were expected in Section 2.3? What exact frequencies were measured?
The frequency spacing expected between the spectrum of the received baseband signal is 122 Hz, i.e the vibration frequency measured by the IMU. The section 2.3.2 has been updated to clarify this point.
- With 6000 rpm about 100Hz of vibrational frequency could be expected. Why is there a discrepancy?
We included the reference of the vibrator and the nominal voltage (1.5 V). Specially, the vibration frequency is dependent of vibrator supply voltage. Thus, the comparison between figure 3.a and 3.b shows the matching between vibration frequency and harmonics spacing of received baseband signal.
- The central peak in Figure 3.a is not explained. It can be assumed that it is the direct-signal. However, a frequency offset of several Hertz is present.
As we previously explained, the frequency offset was corrected in figure 3. This frequency offset was due to the fact that emitter and receiver is not synchronised. Thus, we suppressed the offset to avoid misunderstanding from the readers but we precised in section 2.3.2 that figure 3.a correspond
to synchronised received baseband signal.
The labeling of Figure 3 (a) is cut off.
Your remark has been taken into account
The abbreviation MPSK should be explained.
The abbreviation has been introduced in section 3.1.
Footnote 3 indicates that further experiments with BPSK, QPSK were performed. The results would be of interest for this journal, but they are not shown. Thus, the presented results appear incomplete. It would be interesting for the reader to know if and where these results will be published.
This footnote has been removed due to the fact that experiments mentioned were probably part of an extended version of this article. We provide to reviewer 1 a document describing the mentioned experiments. Thus, the experiments about the BPSK and QPSK are described in section 1 of this document.
Labels (a) and (b) in Figure 4 are partially cut off.
Your remark has been taken into account
Section 4.2 gives a theoretical explanation of a method for micro-movement detection is given. Afterwards simulation results are provided. Why wasn’t the method applied to the measured data from the previous sections?
According to your remark, some explanations of this choice have been added in section 4.2.2. Indeed, the simulation were more convenient than experimental data in regards to controllability and reproductibility.
Please provide references to the list of estimation techniques in section 4.3.1.
According to your remark, we added a reference to [23] (”Spectral Analysis of signals” and an other reference to [31] (”Image and video processing: From Mars to Hollywood with a stop at the hospital - Sparse Modeling” - G. Sapiro)
The authors should provide more explanation behind the “Reconstruction MSE” value in Figure 7.
The metric used for estimation has been specified in section 4.3.3
The graphs in Figure 7 are partially cut off.
Your remark has been taken into account
The conclusions mention the connection to micro-Doppler in radar, especially in passive radar. A relevant study on the measurement of vibrations in wind-turbines using passive radar is presented in the following paper. It also
gives a possible real-world application for micro-Doppler estimation in radio communications.
The reviewer seems to give the authors a reference about passive radar but the reference is not present in MDPI reviewer remarks. From our point of view, the connection between our article and radar and especially passive radar (besides the micro-Doppler effect) are more the tools used to
extract and analyze the received signal than the applications. Indeed, the applications of exploiting micro-Doppler in radio communications seem different. On one hand, the radar domain characterize targets that not necessary emit signals or, at least, does not exploits the transmit signals. On the other hand, the radio communications characterize targets by their emitted signals.

Reviewer 2 Report

This paper presents a model for the Doppler modulation effect, i.e., the effect of complex movement on the received signal, using a geometrical approach. The study focused on micro-Doppler produced by vibrations in radio communications. By leveraging this phenomenon, passive RF sensing could be performed based on communication. The paper also proposes signal processing techniques for detecting micro-Doppler and estimating its parameters. Furthermore, some experiments illustrating the micro-Doppler effect in radio communication are presented. 

The paper is certainly interesting and contributes significantly to the subject of modeling to sensing of micro-Doppler However, it is not easy to read. The paper should be redesigned, and the analytical development should be included as an appendix. It is possible to present a development based on phasor complex method writing, making it easier for the writers to formulate the problem.

Author Response

This paper presents a model for the Doppler modulation effect, i.e., the effect of complex movement on the received signal, using a geometrical approach. The study focused on micro-Doppler produced by vibrations in radio communications. By leveraging this phenomenon, passive RF sensing could be performed based on communication. The paper also proposes signal processing techniques for detecting micro-Doppler and estimating its parameters. Furthermore, some experiments illustrating the micro-Doppler effect in radio communication are presented.
The paper is certainly interesting and contributes significantly to the subject of modeling to sensing of micro- Doppler However, it is not easy to read. The paper should be redesigned, and the analytical development should be
included as an appendix. It is possible to present a development based on phasor complex method writing, making it easier for the writers to formulate the problem.
According to your remark, the analytical development of section 2.2 has been moved in appendix A. Furthermore, the reason that phasor technique was not used is to avoid some misunderstanding between baseband signal and phasor (which corresponds to analytic signal). However, we used analytic signal in the new demonstration to neglect amplitude modulation due to movement.

Reviewer 3 Report

The proposed paper is very interesting.

The topics are relevant both in the field of telecommunications and in the field of radar applications.

Unfortunately, the paper is not linear and the goal of the authors is not always completely clear.

The main improvements I suggest to the author concern the following aspect:

a) An extensive revision of the English language is necessary.

b) A greater linearity in the passage from the hypothesis to the thesis is desirable.

c) Introduction has to better identify the objectives of the paper, the starting points and the methodology used.

Author Response

The proposed paper is very interesting.
The topics are relevant both in the field of telecommunications and in the field of radar applications. Unfortunately, the paper is not linear and the goal of the authors is not always completely clear.
The main improvements I suggest to the author concern the following aspect:
a) An extensive revision of the English language is necessary.
According to your remark, an extension revision of the English language has been performed.
b) A greater linearity in the passage from the hypothesis to the thesis is desirable.
In section 2.2, the hypotheses have been specified and the analytical development of geometrical modeling has been moved in appendix A to increase linearity of the paper.
c) Introduction has to better identify the objectives of the paper, the starting points and the methodology used.
According to your remark, a sentence has been added in the introduction to better explain the objectives of the paper.

Reviewer 4 Report

From modeling to sensing of micro-Doppler in radio communications 

This manuscript described a model for Doppler modulation effect, i.e. the effect of complex movement on the received signal, using a geometrical approach. The technical details are rich, but there is a lack of comparison with state of the art methods, and a description of their application. 

Here are some suggestions for the authors:

1. A comprehensive and concise introduction of the current research is necessary. The problems perplexing the current researchers, as the author described in the paper, should be analyzed elaborately. And what problems or difficulties have you ever solved in your work according to the existing researches? Compared with the existing researches, what improvements and innovations do you have in your work to address the problems? “related work” should be added.

2. The proposed algorithm results is lower than Matching Pursuit in 10dB-20dB. And only Matching Pursuit or Root-MUSIC is not enough, more the-state-of-art methods should be compared.

3. More details of potential applications of the micro-Doppler effect in radio communication context should be described to demonstrate the utility of this finding.

In short, this manuscript seems good idea but shows lack of conclusive results. Therefore, it brings me to this conclusion that the manuscript cannot be published in Remote Sensing.

Author Response

From modeling to sensing of micro-Doppler in radio communications
This manuscript described a model for Doppler modulation effect, i.e. the effect of complex movement on the received signal, using a geometrical approach. The technical details are rich, but there is a lack of comparison with
state of the art methods, and a description of their application.
Here are some suggestions for the authors:
1. A comprehensive and concise introduction of the current research is necessary. The problems perplexing the current researchers, as the author described in the paper, should be analyzed elaborately. And what problems or
difficulties have you ever solved in your work according to the existing researches? Compared with the existing researches, what improvements and innovations do you have in your work to address the problems? “related work”
should be added.
The works present in this paper are relatively new. As we explained during the introduction, few references exploit Doppler effect for passive sensing applications in radio communications domain [13, 14]. To the best of our knowledge, it is the only papers about this subject, limiting thereby the ”relevant work”.
2. The proposed algorithm results is lower than Matching Pursuit in 10dB-20dB. And only Matching Pursuit or Root-MUSIC is not enough, more the-state-of-art methods should be compared.
Effectively, for T=1s, Matching Pursuit in more efficient than our algorithm between 10 to 20 dB. However, our algorithm is more efficient for lower SNR. Furthermore, for T=10s, our algorithm is more efficient than both Root-MUSIC and Matching Pursuit. Concerning your remark about the used of others techniques, we explained in section 4.3.1 the different techniques available. We choose to discard classical spectral estimation because there are too generic and to discard specific estimation techniques because they are the too specific. To the best of our knowledge, we don’t find others references in the literature matching our needs.
3. More details of potential applications of the micro-Doppler effect in radio communication context should be described to demonstrate the utility of this finding.
As we explained previously, passively exploiting Doppler effect in radio communications, especially micro-Doppler is relatively new. The goal of our paper is to present the Doppler modulation phenomenon in radio communications both theoretically and experimentally and to explain how to extract and exploit it. Moreover, in section 5, we propose several applications among: Simulation, Vibration object detection, RF Fingerprinting and Vibration Analysis. We hope that future applications will be proposed by the scientific communities.
In short, this manuscript seems good idea but shows lack of conclusive results. Therefore, it brings me to this conclusion that the manuscript cannot be published in Remote Sensing.
Concerning the conclusive results, we provide both theoretical and experimental results such as {equation (23), figure (3)} and {equation (35), figure (4)}. We hope that taking into account our answers, you could revise your judgment.

Round 2

Reviewer 3 Report

I appreciated the improvements made.

In this version, understanding of the paper has improved.

Corrected the description of the model in the appendix.

There are still some English filings that should be done and that would allow a better understanding of the text

Author Response

I appreciated the improvements made.
In this version, understanding of the paper has improved.
Corrected the description of the model in the appendix.
There are still some English filings that should be done and that would allow a better understanding of the text
An extensive correction of English mistakes in the appendices has been done, specially in appendix A.
